# Experimental observation of classical analogy of topological entanglement entropy

Tian Chen[1], Shihao Zhang [1], Yi Zhang [2], Yulong Liu [1], Su-Peng Kou[3], Houjun Sun[2] & Xiangdong Zhang[1]

Long-range entanglement is an important aspect of the topological orders, so efficient methods to characterize the long-range entanglement are often needed. In this regard, topological entanglement entropy (TEE) is often used for such a purpose but the experimental observation of TEE in a topological order remains a challenge. Here, we propose a scheme to observe TEE in the topological order by constructing specific minimum entropy states (MESs). We then experimentally construct the classical microwave analogs of the MESs and simulate the nontrivial topological order with the TEE in Kitaev toric code, which is in agreement with theoretical predictions. We also experimentally simulate the transition from $Z_2$ topologically ordered state to topologically trivial state.

[1] Key Laboratory of advanced optoelectronic quantum architecture and measurements of Ministry of Education, School of Physics, Beijing Institute of Technology, 100081 Beijing, China. [2] School of Information and Electronics, Beijing Institute of Technology, 100081 Beijing, China. [3] Department of Physics, Beijing Normal University, 100875 Beijing, China. These authors contributed equally: Tian Chen, Shihao Zhang  Correspondence and requests for materials should be addressed to S.-P.K. (email: spkou@bnu.edu.cn) or to H.S. (email: sunhoujun@bit.edu.cn) or to X.Z. (email: zhangxd@bit.edu.cn)

Many physical phenomena can be understood in terms of their topological properties. The presence of topological order in matter is responsible for some fundamental phenomena, such as fractional quantum Hall effect[1], topological spin liquids[2–4], and so on. A topologically ordered phase is an exotic quantum phase that cannot be explained by conventional models based on local order parameters and symmetry breaking[5,6]. Instead, the topological order describes a phase of matter whose correlations satisfy an area law while maintaining long-range entanglement[7,8]. Nontrivial topological properties can exist in topological orders, e.g., topological degeneracy[9], anyons with fractional statistics[10], topologically protected edge states[11], topological entanglement entropy (TEE)[12,13], and so on. Up to now, topological orders have been simulated using different experimental approaches, such as photons[14,15], nuclear spins[16–21], superconducting quantum circuit[22], cold atoms on optical lattice[23]. Furthermore, since topologically ordered phases are robust against local perturbations, they are promising candidates for performing some tasks such as topologically protected quantum computation[24–27]. Thus, characterizing topologically ordered phases and its associated long-range entanglement becomes an important topic in condensed matter physics. An efficient method to detect the long-range entanglement in topological ordered phase is to use TEE[12,13,28,29].

The TEE was introduced by Kitaev and Preskill[12], and Levin and Wen[13] 12 years ago as an unambiguous identifier of topological order within globally entangled ground states. Subsequently, it has been the focus of numerous theoretical and numerical studies[28–33]. It has been demonstrated that TEE can play the crucial role in diagnosing the topological orders and describing their long-range entanglement[28]. Because the TEE always vanishes for the system with trivial topological properties, a nonzero TEE indicates that the system belongs to a topological ordered state. However, these phenomena are only the results of theoretical analyses, they have never been observed in experiments.

In this work, we propose and experimentally demonstrate a scheme to observe the classical analogy of TEE. We demonstrate theoretically that the TEE charactering the topological order for the toric code model can be obtained by constructing specific minimum entropy states (MESs)[34–37]. Furthermore, we construct experimentally the classical microwave analogs of these states, which are mathematically equivalent to their quantum counterparts, and simulate the nontrivial topological order with TEE that is in agreement with the theoretical prediction. Based on this scheme, we also experimentally simulate the transition from $Z_2$ topologically ordered state to topologically trivial phase, which is also consistent with the theoretical results.

## Results

**Theoretical scheme for measuring TEE based on MESs.** The two-dimensional toric code model is firstly proposed by Kitaev, which can exhibit the $Z_2$ topological order in the model without external fields[24,25]. The model Hamiltonian is

$$H_0 = -\sum_s A_s - \sum_p B_p, \qquad (1)$$

where $A_s = \prod_{i \in s} \sigma_i^x$ and $B_p = \prod_{i \in p} \sigma_i^z$, the subscripts $s$ and $p$ represent the vertices and plaquettes of a square lattice, respectively. The schematic diagram for the toric code model is illustrated in Fig. 1a. The operators $A_S$ and $B_p$ containing four-body interaction of local spin are represented by yellow cross and red square, respectively. The operator $\sigma_i^{x(z)}$ in the Hamiltonian are the Pauli

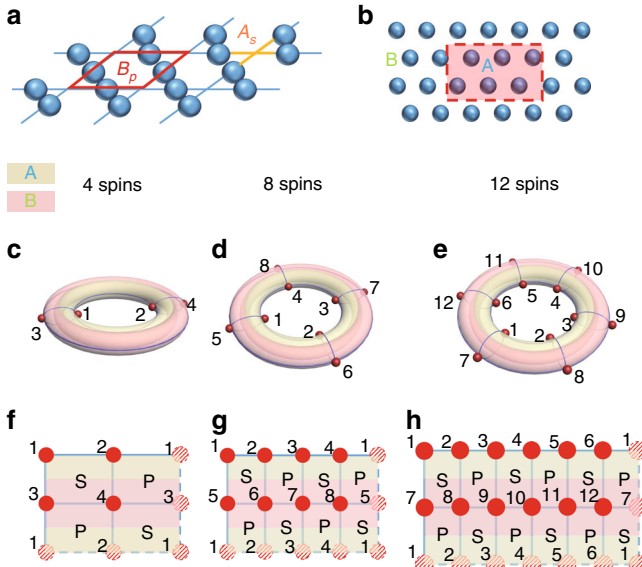

**Fig. 1** Separation of the toric code model into two subsystems. **a** the schematic diagram of the toric code model. A blue sphere stands for a spin, $A_s$ and $B_p$ represent two kinds of four-body interactions in the Hamiltonian $H_0$. **b** the total system is divided into subsystems A and B. The boundary is shown by red dashed line. In **c–e** the yellow region is the subsystem A, and the pink region is the subsystem B. Each red sphere represents one spin. **c–e** the torus geometry for square lattices with 4, 8, and 12 spins. The periodic boundary conditions are taken. **f–h** the unfolded forms of the corresponding torus in **c–e**, respectively. The shadow spheres are drawn to show the periodic boundary condition. Here, the operator $S$ represents the operator $A_s$ and $P$ represents the operator $B_p$. Two disconnected boundaries exist between subsystem A and B, and the lengths for each boundary between subsystem A and B in **f–h** are 2, 4, and 6, respectively

operators. Many methods have been proposed to evaluate the nontrivial topological order in this model[12,13,18–20,28,29,30–32].

Since it has been recognized that the topological property of the model is directly associated with the underlying entanglement in its ground state $|\varphi_g\rangle_i (i = 1, 2, …, n)$, one efficient method of characterizing topological order is to use the TEE obtained from ground states of the model. The TEE characterizing the long-range entanglement in the system is extracted from von Neumann entropies for different ground states. To evaluate the von Neumann entropy of the ground state, we need to divide the model into two subsystems (A and B) and create a boundary between these two subsystems as shown in Fig. 1b. The boundary between subsystems A and B is represented by red dashed line. The von Neumann entropy for the subsystem A is expressed as $S_A = -\mathrm{Tr}\rho_A \log \rho_A$, where $\rho_A = \mathrm{Tr}_B(|\varphi_g\rangle_{ii}\langle\varphi_g|)$ is the reduced density matrix. As addressed in refs. [12,13,28], the von Neumann entropy for the subset A of the lattice is a linear function as the length of the smooth boundary $L_x$, that is, $S_A = \alpha L_x - m\gamma + \cdots$. The coefficient $\alpha$ resulting from short wavelength modes near the boundary is non-universal, the ellipsis represents terms that vanish in the limit $L_x \rightarrow \infty$, $m$ is the number of disconnected boundaries and $\gamma$ is the TEE that is a universal additive constant characterizing the long-range entanglement in the ground state. In general, the $\gamma$ is defined to be $\ln D$, where $D = \sqrt{\sum_a d_a^2}$ is the total quantum dimension of the medium and the sum is over all the superselection sectors of the medium, $d_a$ is the quantum dimension of a quasi-particle with charge $a$. For the $Z_2$ topological order, the TEE is $\gamma = \ln 2 = 0.6932$. The TEE has

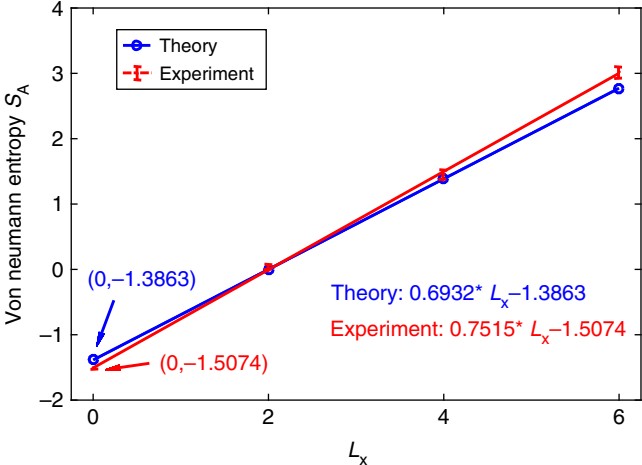

**Fig. 2** The von Neumann entropies for the subsystem A of MESs $|\psi\rangle_4$, $|\psi\rangle_8$, and $|\psi\rangle_{12}$. Blue solid line is obtained from theoretic results, and red dotted dashed line represents experimental results. Error bars are defined as s.d. Source data are provided as a Source Data file

been calculated numerically in the previous study on ground states of the model using density matrix renormalization group (DMRG) method[28].

Because it has been demonstrated that the toric code model is equivalent to a square lattice in terms of studying the TEE[28], we consider the equivalent square lattice as shown in Fig. 1c–h, where the operators $\sigma_x$ and $\sigma_z$ occupy the vertices instead of bounds. In fact, different boundary conditions can be chosen to analyze the physical characteristics of the system. However, if the boundary condition is not chosen appropriately, the non-universal term from sharp corner affects the entanglement entropy for the lattice and the TEE is not directly obtained[33]. To avoid this sharp corner contribution, we choose the periodic boundary condition in the lattice and the system displays the torus geometry Fig. 1c–e. The schematic representations in Fig. 1f–h are the unfolded forms of the corresponding torus geometry in Fig. 1c–e. The operators $A_s$ and $B_p$ in the toric code model now sit on alternating plaquettes in the equivalent square lattice, which are labeled as $S$ and $P$ operators, respectively. As shown in Fig. 1c–h, we consider the square lattice involving 4, 8 and 12 spins, respectively. One subsystem covered by the yellow color is the subsystem A, and the other with the pink color belongs to the subsystem B. Since we choose the periodic boundary condition in the system, there are two disconnected boundaries between subsystem A and B. For the lattice with 4, 8, and 12 spins, the length of each disconnected boundary are $L_x = 2$, 4 and 6, respectively. The ground state $|\varphi_g\rangle_i$ of the square lattice is given by $S|\varphi_g\rangle_i = |\varphi_g\rangle_i$ and $P|\varphi_g\rangle_i = |\varphi_g\rangle_i$ for all plaquettes. Due to the periodic boundary condition in the square lattice, there exist four degenerated ground states. The TEE for the lattice can be extracted from MESs, which are equal superposition of two degenerated ground states having the same parity of winding number along one cycle of torus[34]. Details of obtaining MESs are presented in Methods for the square lattice with 4 spins and in Supplementary Note 1 for the square lattices with 8- and 12-spins. When we choose the non-contractible cut in the torus geometry (Fig. 1c–e), the TEE can be obtained as a constant (not $L_x$-dependence) in the entanglement entropy for MES[34]. This TEE value can be extracted from entanglement entropies for different system sizes. The proof of correspondence between the square lattice and the toric code model has been given in Supplementary Note 2. The obtained MESs for the square lattices

with 4, 8, and 12 spins ($|\psi\rangle_4$, $|\psi\rangle_8$, and $|\psi\rangle_{12}$) are shown below.

$$|\psi\rangle_4 = 1/2(|0000\rangle + |1111\rangle + |0011\rangle + |1100\rangle), \qquad (2)$$

$$\begin{aligned}
|\psi\rangle_8 = \tfrac{1}{4}(&|00000000\rangle + |00110011\rangle + |01010101\rangle + |01100110\rangle \\
&+ |10011001\rangle + |10101010\rangle + |11001100\rangle + |11111111\rangle \\
&+ |00001111\rangle + |00111100\rangle + |01011010\rangle + |01101001\rangle \\
&+ |10010110\rangle + |10100101\rangle + |11000011\rangle + |11110000\rangle
\end{aligned}$$
$$(3)$$

$$\begin{aligned}
|\psi\rangle_{12} = \tfrac{1}{8}(&|000000000000\rangle + |000011000011\rangle \\
&+ |000101000101\rangle + |000110000110\rangle + |001001001001\rangle \\
&+ |001010001010\rangle + |001100001100\rangle + |001111001111\rangle \\
&+ |010001010001\rangle + |010010010010\rangle + |010100010100\rangle \\
&+ |010111010111\rangle + |011000011000\rangle + |011011011011\rangle \\
&+ |011101011101\rangle + |011110011110\rangle + |100001100001\rangle \\
&+ |100010100010\rangle + |100100100100\rangle + |100111100111\rangle \\
&+ |101000101000\rangle + |101011101011\rangle + |101101101101\rangle \\
&+ |101110101110\rangle + |110000110000\rangle + |110011110011\rangle \\
&+ |110101110101\rangle + |110110110110\rangle + |111001111001\rangle \\
&+ |111010111010\rangle + |111100111100\rangle + |111111111111\rangle \\
&+ |000000111111\rangle + |000011111100\rangle + |000101111010\rangle \\
&+ |000110111001\rangle + |001001110110\rangle + |001010110101\rangle \\
&+ |001100110011\rangle + |001111110000\rangle + |010001101110\rangle \\
&+ |010010101101\rangle + |010100101011\rangle + |010111101000\rangle \\
&+ |011000100111\rangle + |011011100100\rangle + |011101100010\rangle \\
&+ |011110100001\rangle + |100001011110\rangle + |100010011101\rangle \\
&+ |100100011011\rangle + |100111011000\rangle + |101000010111\rangle \\
&+ |101011010100\rangle + |101101010010\rangle + |101110010001\rangle \\
&+ |110000001111\rangle + |110011001100\rangle + |110101001010\rangle \\
&+ |110110001001\rangle + |111001000110\rangle + |111010000101\rangle \\
&+ |111100000011\rangle + |111111000000\rangle.
\end{aligned}$$
$$(4)$$

As the explicit forms of MESs $|\psi\rangle_4$, $|\psi\rangle_8$, and $|\psi\rangle_{12}$ have been obtained, we get the reduced density matrix $\rho_A$ for the subsystem A (yellow regions in Fig. 1c–e) and present the von Neumann entropies $S_A = -\mathrm{Tr}\rho_A \log\rho_A$ for these states in Fig. 2 (blue circles). We can use a line (blue line) to connect these three entropy values for different lengths of boundary $L_x$. Following the linear relation between the von Neumann entropy $S_A = \alpha L_x - m\gamma + \&hellipsis;$ and $L_x$, we extend the line to $L_x = 0$, and extract the TEE value $\gamma = \ln 2 = 0.6932$, which shows the efficiency of our method. In the following, we explore the experimental simulation of these phenomena.

**Experimental observation of classical analogy of TEE.** According to the above theoretical analysis, a direct experimental scheme to observe the topological order is to use a quantum-many body system[18–21]. However, it is very difficult to experimentally realize such a scheme, e.g., the fidelity of the 12-qubit states recently prepared on the IBM quantum computer is lower than 0.44[38], yet a 12-qubit state with a high fidelity needs to be created to complete the above experiment for observing TEE. Here, we propose a microwave experiment scheme to observe the classical analogy of the nontrivial topological order. Although some specific quantum phenomena have been simulated using

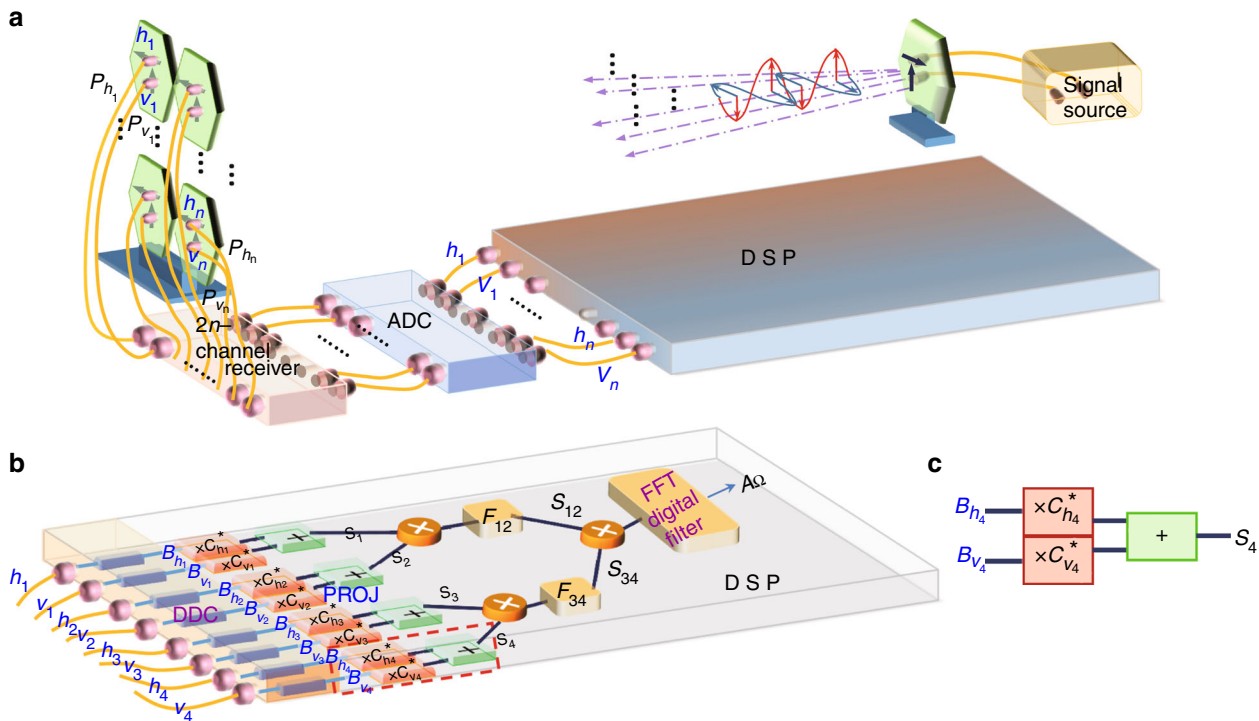

**Fig. 3** The experimental set-up to construct and characterize the CMESs as the analogy of quantum MESs. **a** The overall designed classical microwave signal processing system, including transmitting and receiving dual-polarized antennas, a $2n$-channel receiver array, an analog-to-digital converter (ADC) and a programmable DSP module. **b** An example of the work flow to construct the 4-cebit analog $|\psi_4^{cl}\rangle$ in the DSP module. All eight signals in the channels $\{h_1, ..., v_4\}$ fed into the DSP are arranged by a digital down-converter (DDC) to have their own frequencies as planned. After a measuring process through the PROJ part followed by several mixing and filtering processes for multiplex signals, the desired signal component $A_\Omega$ is selected by the FFT-based digital filter, and we collect its complex amplitude as described in Eq. (5). The red dashed box highlights a unit of the PROJ part as depicted in **c**, including two multiplications and an adder, which can be tuned conveniently to perform desired projective measurement settings

classical microwaves[39], optical beams[40–47], electronic signals[48] or hybrid optical-electrical systems[49,50], the long-range correlated characteristics in the topological order have never been explored. By establishing a mapping between the detection of intensities of classical signals from an appropriate designed circuit and the correlation measurements in the quantum photonic experiments, here we implement a series of microwave experiments with a different number of receiving antennas and signal channels to simulate various MESs. Thus, our work opens up a window to study the topological phase and transition of topological orders. Our experimental setup is presented in Fig. 3a.

We use a dual-polarized antenna to transmit microwave signals with vertical and horizontal polarizations to $n$ distant dual-polarized receiving antennas. The numbers of the receiving antenna arrays and associated signal channels correspond to the qubit number in the desired states. Each dual-polarized receiving antenna includes two channels $\{h_i, v_i\}$, and the induced electric voltage signals in the $h_i$ ($v_i$) channel are denoted as $P_{h_i}(P_{v_i})(i = 1, 2, \ldots, n)$. Then these multiplex signals pass through a $2n$-channel receiver for frequency down-conversion, and subsequently an analog-to-digital converter (ADC) for further digital signal processing (DSP). In the programmable DSP module, we can design a specific process for constructing and measuring the desired classical analogy states.

For example, in order to construct the corresponding form of the 4-qubit state $|\psi\rangle_4$ in Eq. (2), we arrange eight signals in the set of channels $\{h_1, v_1, h_2, v_2, h_3, v_3, h_4, v_4\}$ fed into the DSP module to successively pass through a digital down-converter (DDC), the projection part (PROJ) for realizing projective measurements and a series of mixing and filtering processes as shown in Fig. 3b

(details are shown in Methods). By assuming $P_{h_i} = P_{v_i} = P_i(i = 1, 2, 3, 4)$ under the far-field approximation condition, the corresponding complex amplitude $A_\Omega$ of the final FFT-filtered signal in the frequency domain is expressed as

$$
\begin{aligned}
A_\Omega \propto P_1 P_2 P_3 P_4 \cdot \Big[ & \big(\mathbf{e}_{\mathbf{m}_1}|h_1\big)\big(\mathbf{e}_{\mathbf{m}_2}|h_2\big)\big(\mathbf{e}_{\mathbf{m}_3}|h_3\big)\big(\mathbf{e}_{\mathbf{m}_4}|h_4\big) \\
& + \big(\mathbf{e}_{\mathbf{m}_1}|h_1\big)\big(\mathbf{e}_{\mathbf{m}_2}|h_2\big)\big(\mathbf{e}_{\mathbf{m}_3}|v_3\big)\big(\mathbf{e}_{\mathbf{m}_4}|v_4\big) \\
& + \big(\mathbf{e}_{\mathbf{m}_1}|v_1\big)\big(\mathbf{e}_{\mathbf{m}_2}|v_2\big)\big(\mathbf{e}_{\mathbf{m}_3}|h_3\big)\big(\mathbf{e}_{\mathbf{m}_4}|h_4\big) \\
& + \big(\mathbf{e}_{\mathbf{m}_1}|v_1\big)\big(\mathbf{e}_{\mathbf{m}_2}|v_2\big)\big(\mathbf{e}_{\mathbf{m}_3}|v_3\big)\big(\mathbf{e}_{\mathbf{m}_4}|v_4\big) \Big],
\end{aligned}
\tag{5}
$$

where $\big(\mathbf{e}_{\mathbf{m}_i}| = \big(c_{h_i}^*, c_{v_i}^*\big)$ is the conjugate transpose for $\big|e_{m_i}\big\rangle = \big(c_{h_i}, c_{v_i}\big)^{\mathrm{T}}$ as the projective measurement basis arranged in the PROJ, and $|h_i\rangle = (1, 0)_i^{\mathrm{T}}$ $\big(|v_i\rangle = (0, 1)_i^{\mathrm{T}}\big)$ represents the horizontal (vertical) unit amplitude signal in the $h_i(v_i)$ channel. Here the parentheses notation (parent (|and thesis|)) has been used to describe the cebit states[46,47,49,50]. The cebit represents the vector form of a signal pair as the classical counterpart of a single-qubit quantum state, and these cebits form an inner product space where the inner product is given by parentheses (|)[46,47]. Thus, the signal amplitude in Eq. (5) can be rewritten in the form $A_\Omega \propto \big(\mathbf{e}_{\mathbf{m}_1}|\big(\mathbf{e}_{\mathbf{m}_2}|\big(\mathbf{e}_{\mathbf{m}_3}|\big(\mathbf{e}_{\mathbf{m}_4}|\psi_4^{cl}\big)$, and the $|\psi_4^{cl}\rangle$ is

$$
\begin{aligned}
|\psi_4^{cl}\rangle = \tfrac{1}{2} [&|h_1\rangle|h_2\rangle|h_3\rangle|h_4\rangle + |h_1\rangle|h_2\rangle|v_3\rangle|v_4\rangle \\
&+ |v_1\rangle|v_2\rangle|h_3\rangle|h_4\rangle + |v_1\rangle|v_2\rangle|v_3\rangle|v_4\rangle],
\end{aligned}
\tag{6}
$$

which is the microwave analog of the 4-qubit MES $|\psi\rangle_4$. Such a

form is called 4-cebit classical minimal entropy states (CMES) composed of the combination of the basis $|h_i\rangle$ and $|v_i\rangle$, and the identification of the term cebit is specifically introduced in Supplementary Note 3. In short, after the classical analogy of the projective measurement, we construct specific mixing and filtering processes for signals by analogy with post-selected fusion operations on photons in quantum experimental settings[51,52], and the recorded data $|A_\Omega|^2$ are similar to those multi-fold coincidence events registered in the quantum set-up[51,53–55] at a purely formal level. Such a way is also similar to our previous studies on classical microwave graph states[56], which are classical analogies of quantum graph states.

The form $|\psi_4^{cl}\rangle$ in Eq. (6) is the classical correspondence with $|\psi\rangle_4$, which can be quantified by a traditional state tomography method that has been used to characterize quantum states[57]. Here, a Hermitian and positive semi-definite matrix $\rho_4^{cl}$ with trace 1 corresponding to $|\psi_4^{cl}\rangle\langle\psi_4^{cl}|$ is introduced by analogy with the density matrix of a quantum state. In the field of quantum information, the fidelity is commonly used to judge the quality of a produced state compared with the desired one. Similarly, here we use this notion to measure the degree of similarity between our experimental simulation results $\hat{\rho}_4^{cl}$ and the target analogy state $|\psi_4^{cl}\rangle$, which can be quantified as $\langle\psi_4^{cl}|\hat{\rho}_4^{cl}|\psi_4^{cl}\rangle$ similar to those used in quantum experiments[51–57]. From the projective measurement data, we obtain its fidelity of $0.9977 \pm 0.0009$ in the present case, which reflects good reliability of our experimental simulation. The detailed descriptions of the experiment results and the reconstructed method for the density matrix are given in Supplementary Note 3.

The experimental setup for the 8- and 12-cebit CMESs can be obtained by a modified extension of the 4-cebit scheme in Fig. 3b. The advantages of the present microwave scheme, including good controllability and reliability, make it convenient to be expanded to construct larger CMESs with more cebits. For the 8-cebit scheme, we need 8 dual-polarized receiving antennas and a 16-channel receiver followed by an ADC. Correspondingly, 12 dual-polarized receiving antennas and a 24-channel receiver are needed for the 12-cebit scheme. The detailed experimental setups for 8- and 12-cebit CMESs are provided in Supplementary Note 4. Based on these experimental setups, the corresponding classical analogy of $|\psi\rangle_8$ and $|\psi\rangle_{12}$ can be obtained. Similar to the verification of the 4-cebit CMES, we can also judge the qualities of the 8- and 12-cebit cases by verifying their reconstructed density matrices.

However, the state tomography technique seems experimentally prohibitive for dealing with the 8− and 12-cebit CMESs, since the number of measurement settings required grows exponentially with the number of qubits (cebits). Fortunately, the low rank property of the desired density matrices allows us to employ compressed sensing method to reduce the number of measurements dramatically. The theory of compressed state tomography[58] claims that if the targeted density matrices are low-rank, then one can stably reconstruct these matrices from highly incomplete Pauli measurements via some effective convex recovery procedures. The detailed compressed sensing method for state tomography is included in Supplementary Note 5. Based on this approach, we obtain the fidelity of $0.9855 \pm 0.0045$ and $0.9702 \pm 0.0056$ for 8- and 12-cebit cases, respectively. The high fidelity of our experimental results, demonstrates good reliability of our scheme in constructing CMESs, and also has numerous advantages for exhibiting the topological order with the extraction of TEE from these states.

From the above experimental data of density matrices, we can obtain the corresponding von Neumann entropies. The results for 4-, 8-, and 12-cebit cases are shown by red dotted line in Fig. 2.

Note the two disconnected boundaries between subsystems A and B, from the experimental results we obtain the TEE value as $\gamma = 0.7537$, which is basically identical with the prediction of quantum theory. The experimental errors can be further reduced by improving the efficiency of the microwave signal collection and the control accuracy of the electrical signal. This means that we have provided a good experimental platform for demonstrating the classical analogy of topological order with the TEE. Based on such an experimental platform, we can exhibit not only the nontrivial topological order but also the transition from $Z_2$ topological order to trivial spin polarized phase in Kitaev toric code model. In the following, we study the experimental simulation on such a transition based on CMESs.

**Classical analogy of transition from $Z_2$ order to trivial phase.** The toric code model undergoes the transition from topologically ordered phase to topologically trivial phase with the change of external fields[28]. The Hamiltonian for the toric code model with external fields is

$$H_t = H_0 + H_1, \tag{7}$$

with

$$H_1 = -g \sum_{i=1}^{n} (\sigma_x^i + \sigma_z^i), \tag{8}$$

where the coefficient $g$ represents the strength of symmetrically external fields. Similar to the lattice without external fields, the study of TEE in the toric code model is equivalent to the square lattice when external fields exist. Details of equivalence between the toric code model and the square lattice in such a case has been provided in Supplementary Note 7. Therefore, in the following we study the transition from $Z_2$ topological order to trivial phase based on the equivalent square lattice with the change of the external field. Since the topological order can be described from the TEE associated with the MESs, we can observe the transition from topologically ordered phase to trivial phase if we obtain the MESs of the model belonging to different phases. However, there does not exist the analytic expression for the ground state of the Hamiltonian in Eq. (7). By using numerical methods, we can get the ground states for the model with external fields and obtain the MESs based on these ground states. We take the system involving 4 spins as an example, when the system belongs to $Z_2$ topological order, there exists four nearly degenerated ground states. We do the linear combination for these four nearly degenerated ground states and obtain the MES. When the strength $g$ equals to 0.1, the system belongs to $Z_2$ topological order and the MES is

$$|\Xi\rangle_4 = a_{4,1}|0000\rangle + a_{4,2}|0001\rangle + \ldots + a_{4,16}|1111\rangle. \tag{9}$$

Here, the coefficients $a_{4,i}(i=1,\ldots,16)$ are $a_{4,1} = -0.5386$, $a_{4,4} = a_{4,13} = 0.5072$, $a_{4,6} = a_{4,11} = 0.006$, $a_{4,7} = a_{4,10} = -0.0157$, $a_{4,16} = -0.4415$, and other coefficients take zero. When the strength $g$ equals to 0.2, the system remains in the topologically order phase, and we can obtain the MES of system as above. The calculation details can be found in Supplementary Note 6. When the coefficient $g$ increases to a certain value, the transition from $Z_2$ topological order to trivial phase emerges in the system. In this case, there are no nearly degenerated ground states and the topological properties can be revealed from its unique ground state. In order to describe the topological properties of system with the external fields, we need to obtain the corresponding ground states of systems. Here, we provide the ground state when the strength $g$ equals to 10:

$$\left|\Phi_g\right\rangle_4 = a_{4,1}|0000\rangle + a_{4,2}|0001\rangle + \ldots + a_{4,16}|1111\rangle, \tag{10}$$

with $a_{4,1} = 0.7408$, $a_{4,2} = a_{4,3} = a_{4,5} = a_{4,9} = 0.2883$, $a_{4,4} = a_{4,6} =$

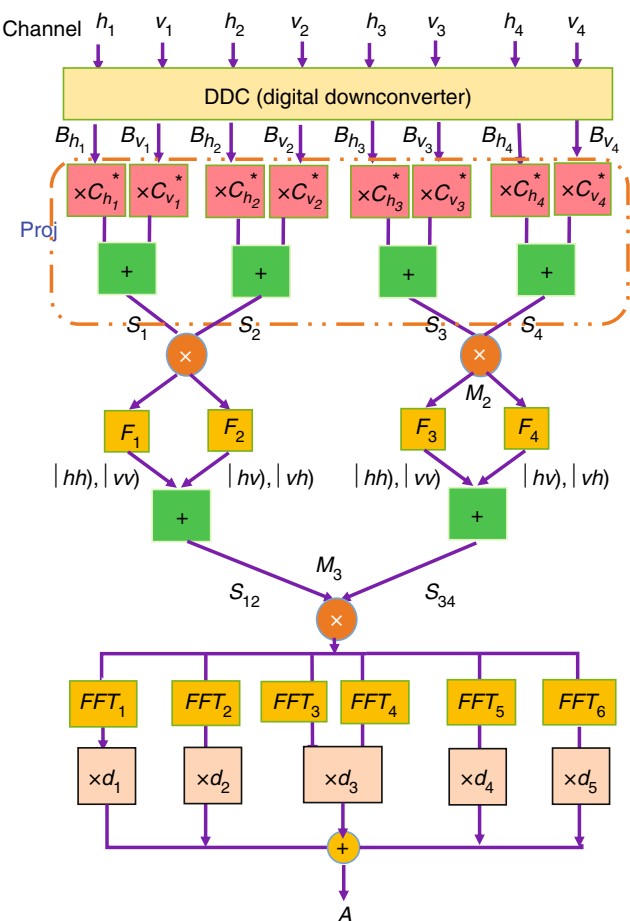

**Fig. 4** The designed work flow to obtain the classical analogy of the state $|\Xi\rangle_4$ in the DSP module. All eight signals in the channels $\{h_1, v_1, ..., h_4, v_4\}$ fed into the DSP are digitally down-converted to their own frequencies as planned. After a measuring process through the PROJ part followed by several mixing and filtering processes for multiplex signals, the desired frequency components selected by a collections of FFT-based digital filters $\{FFT_1, ..., FFT_6\}$ are adjusted by five modulators $\{d_1, ..., d_5\}$, respectively, and we collect their joint complex amplitude denoted as $A$. Also, this procedure can be conveniently adjusted to simulate the state $|\Phi_g\rangle_4$

$a_{4,7} = a_{4,10} = a_{4,11} = a_{4,13} = 0.1303$, $a_{4,8} = a_{4,12} = a_{4,14} = a_{4,15} = 0.0615$, and $a_{4,16} = 0.0412$. The explicit forms of ground states with the strength $g = 0.9, 1, 2, 5$ and the MESs for the system involving 8 spins are also given in Supplementary Note 6.

The classical analogy of these MESs and ground states can be obtained in the microwave experiment as shown in Fig. 3a. Unlike the case without the external fields, the DSP module needs to be redesigned. Figure 4 shows the experimental setup of the DSP module for the 4-cebit cases with a small $g(=0.1, 0.2)$, which is similar to the scheme presented in Fig. 3b. Firstly, the eight signals from the set of channels $\{h_1, v_1, h_2, v_2, h_3, v_3, h_4, v_4\}$ are injected into the DSP module. After the digital down-conversion and PROJ, we obtain four recombined signals $\{S_1(t), S_2(t), S_3(t), S_4(t)\}$ and then send them in pair into two multipliers $\{M_1, M_2\}$ for mixing processes, respectively.

In order to simulate the state $|\Xi\rangle_4$, we let the output mixing signals pass through four appropriate filters $\{F_1, F_2, F_3, F_4\}$ and two adders in parallel to output two summed signals $S_{12}$ and $S_{34}$, followed by a multiplier $M_3$ for mixing and a collection of FFT-based digital filters $\{FFT_1, ..., FFT_6\}$. When the required frequency components are filtered, we use five modulators $\{d_1, ..., d_5\}$ to adjust the amplitudes of these components respectively

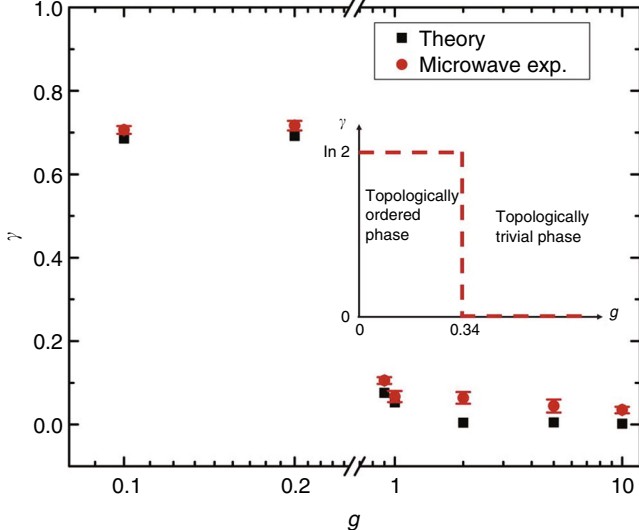

**Fig. 5** The TEE for the square lattice with strength of external fields $g$. Red circles represent results from the microwave experiments. Black squares are theoretic results of TEE obtained from the corresponding MESs. Inset: when the model belongs to $Z_2$ topological order, the long-range entanglement in the MES indicated as the TEE is $\gamma = \ln 2$; the model belongs to the trivial phase when the external field becomes strong and the TEE is $\gamma = 0$. Error bars are defined as s.d. Source data are provided as a Source Data file

to obtain the corresponding expanded coefficients in Eq. (9). The final recorded amplitudes $A$ in the frequency domain corresponds to the projective measurement results of $|\Xi\rangle_4$. For the case with a large $g(= 10, 5, 2, 1, 0.9)$, only certain mixing, filtering and amplitude modulation in the DSP module are adjusted in a similar way to those with small $g$, and the final recorded amplitudes $A$ in the frequency domain corresponds to the projective measurement results of $|\Phi_g\rangle_4$ (details are shown in Supplementary Fig. 16). This means that we have experimentally obtained the classical analogy of the corresponding ground states in Eq. (10).

The experimental setup for obtaining the classical analogy of the 8-qubit state in the cases with a small $g(= 0.1, 0.2)$ or a large $g$ $(= 10, 5, 2, 1, 0.9)$ can be regarded as an extension of the corresponding 4-cebit case with external fields. That is to say, the experimental elements in Fig. 4 need to be doubled with additional processes to achieve desired 8-cebit classical analogs of the states $|\Xi\rangle_8$ and $|\Phi_g\rangle_8$. The designed processes for these 4– and 8-cebit cases with detailed descriptions are presented in Supplementary Note 8.

After obtained experimental results of the classical analogy of density matrices for various $g$, we get the reduced density matrix $\rho_A$ for the subsystem A and calculate the von Neumann entropies $S_A = -\mathrm{Tr}\rho_A \log \rho_A$ for these states. Then we can extract the TEE at various external fields. Red circles in Fig. 5 are experimental results for $g = 0.1, 0.2, 0.9, 1, 2, 5, 10$, and black squares represent theoretical calculation results. The agreement between the experimental results and theoretical calculations is verified. Both experimental and theoretical results show that when $g$ is small ($g = 0.1, 0.2$), the TEE $\gamma$ has larger values which reveal the $Z_2$ topological order; while, when $g$ is large ($g = 0.9, 1, 2, 5, 10$), the TEE $\gamma$ is close to zero which indicates the trivial phase. This means that the transition from topologically ordered phase to topologically trivial phase emerges with the increase of $g$. These phenomena disclosed by our experiments coincides with the exact numerical study of TEE based on DMRG in ref. [28].

## Discussion

In this work, we have exploited multiple microwave beams and signal processing techniques to observe an analogy of the TEE extracted from the 4−, 8−, and 12-qubit systems. In fact, in the past years many simulations (or emulations) of quantum multipartite systems have been performed by using multiple classical optical beams[46,47,49,50], microwaves[39] or electronic signals[48] with in-depth discussions of the cost of classical resources. Inspired by this area of research, we have investigated how to employ a specific classical system to observe an analogy of topological phases in the toric code model and also the transition from $Z_2$ topologically ordered state to topologically trivial phase, and thus obtained enlightening results in agreement with theoretical predictions.

However, the difference between a quantum simulator and our classical analogy is the scaling behavior of the resources required in simulating the target quantum many-body system. A well-designed genuine quantum system (e.g. superconducting quantum circuits) usually owns good scalability in terms of the resources used[59,60], while a classical analog system does not for most cases. For example, for the case of the analogs of the 8-qubit states $|\Xi\rangle_8$ and $|\Phi_g\rangle_8$ with external fields, each superposed term in the target state is identified with a frequency component and selected by the use of an associated FFT-based filter followed by a modulator. Thus, the total number of filters required scales as $2^n$ with the qubit number $n$. That is to say, the whole bandwidth required for the filtering process would grow exponentially with the number of qubits in a general state to be simulated, similar to the conclusions drawn in previous work[48]. In contrast to these analogies of general states, it is also interesting to note for the analogy of certain specific states (e.g. the 4-, 8-, 12-cebit CMES), the amount of employed classical resources can be reduced to some extent by appropriate designs. In particular, in our design the numbers of receiving antennas, signal channels $\{h_i, v_i\}$ fed into the DSP module, and digitally down-converted signals $\{B_{h_i}(t), B_{v_i}(t)\}$ grow linearly with the number of cebits (details can be found in Fig. 3b, Supplementary Figs. 9 and 10), while the number of FFT-based filters at the bottom of each designed circuit in the DSP module usually depends on the number of superposed terms in the target state.

In summary, we have proposed a way to simulate topological order with TEE based on classical analogies of certain quantum states. We have verified theoretically that TEE charactering the topological order for the toric code model can be obtained by constructing the minimal entropy states. The corresponding microwave experiments have been performed and the CMESs have been observed. Based on these classical analogs, we have obtained the von Neumann entropy and the nontrivial topological order, which are agreement with the theoretical predictions. Moreover, we have extended this scheme to the case with the external fields and simulated experimentally the transition from topologically ordered phase to topologically trivial phase, which is also identical with the theoretical results. Our studies not only represent an important advance in the study of TEE, but also open up an avenue to explore some intriguing topological properties based on high-fidelity microwave analogies.

## Methods

**Finding MESs for the square lattice**. To extract the universal constant TEE for the square lattice, we need to find the MESs which are a linear combination of degenerated ground states. For the square lattice with 4 spins, there exist four degenerated ground states. By using the numerical diagonalization to the system Hamiltonian, we can obtain them. The Hamiltonian for the square lattice with 4 spins $H_{0,4}$ is expressed as

$$H_{0,4} = -2\big(\sigma_x^1\sigma_x^2\sigma_x^3\sigma_x^4 + \sigma_z^1\sigma_z^2\sigma_z^3\sigma_z^4\big) \tag{11}$$

Corresponding to such a system, there are four degenerated ground states,

$$\big|\varphi_{g,1}\big\rangle_4 = \tfrac{1}{\sqrt{2}}(|0000\rangle + |1111\rangle),\ \big|\varphi_{g,2}\big\rangle_4 = \tfrac{1}{\sqrt{2}}(|0011\rangle + |1100\rangle),\ \big|\varphi_{g,3}\big\rangle_4$$
$$= \tfrac{1}{\sqrt{2}}(|1010\rangle + |0101\rangle),\ \big|\varphi_{g,4}\big\rangle_4 = \tfrac{1}{\sqrt{2}}(|1001\rangle + |0110\rangle). \tag{12}$$

Now, we define the string operators as

$$\widetilde{F}_x = \sigma_x^1\sigma_x^2,\ \widetilde{F}_y = \sigma_x^1\sigma_x^3,\ \widetilde{T}_x = \sigma_z^1\sigma_z^2,\ \widetilde{T}_y = \sigma_z^1\sigma_z^3. \tag{13}$$

The operators $\widetilde{F}_x$ and $\widetilde{F}_y$ represent the magnetic charge loop operators, the operators $\widetilde{T}_x$ and $\widetilde{T}_y$ are the electric charge loop operators. Applying these string operators to the degenerated ground states, we have

$$
\begin{aligned}
\widetilde{F}_x\big|\varphi_{g,1}\big\rangle_4 &= \big|\varphi_{g,2}\big\rangle_4, \widetilde{F}_x\big|\varphi_{g,2}\big\rangle_4 = \big|\varphi_{g,1}\big\rangle_4, \widetilde{F}_x\big|\varphi_{g,3}\big\rangle_4 = \big|\varphi_{g,4}\big\rangle_4, \widetilde{F}_x\big|\varphi_{g,4}\big\rangle_4 \\
&= \big|\varphi_{g,3}\big\rangle_4, \widetilde{F}_y\big|\varphi_{g,1}\big\rangle_4 = \big|\varphi_{g,3}\big\rangle_4, \widetilde{F}_y\big|\varphi_{g,2}\big\rangle_4 = \big|\varphi_{g,4}\big\rangle_4, \widetilde{F}_y\big|\varphi_{g,3}\big\rangle_4 \\
&= \big|\varphi_{g,1}\big\rangle_4, \widetilde{F}_y\big|\varphi_{g,4}\big\rangle_4 = \big|\varphi_{g,2}\big\rangle_4, \widetilde{T}_x\big|\varphi_{g,1}\big\rangle_4 = \big|\varphi_{g,1}\big\rangle_4, \widetilde{T}_x\big|\varphi_{g,2}\big\rangle_4 \\
&= \big|\varphi_{g,2}\big\rangle_4, \widetilde{T}_x\big|\varphi_{g,3}\big\rangle_4 = -\big|\varphi_{g,3}\big\rangle_4, \widetilde{T}_x\big|\varphi_{g,4}\big\rangle_4 = -\big|\varphi_{g,4}\big\rangle_4, \widetilde{T}_y\big|\varphi_{g,1}\big\rangle_4 \\
&= \big|\varphi_{g,1}\big\rangle_4, \widetilde{T}_y\big|\varphi_{g,2}\big\rangle_4 = -\big|\varphi_{g,2}\big\rangle_4, \widetilde{T}_y\big|\varphi_{g,3}\big\rangle_4 = \big|\varphi_{g,3}\big\rangle_4, \widetilde{T}_y\big|\varphi_{g,4}\big\rangle_4 = -\big|\varphi_{g,4}\big\rangle_4.
\end{aligned} \tag{14}
$$

The MESs for the square lattice with 4 spins can be obtained by the linear combination of these degenerated ground states[30], which are expressed as:

$$|\Xi_1\rangle = \tfrac{1}{\sqrt{2}}\left(\big|\varphi_{g,1}\big\rangle_4 + \big|\varphi_{g,2}\big\rangle_4\right),\ |\Xi_2\rangle = \tfrac{1}{\sqrt{2}}\left(\big|\varphi_{g,1}\big\rangle_4 - \big|\varphi_{g,2}\big\rangle_4\right),\ |\Xi_3\rangle$$
$$= \tfrac{1}{\sqrt{2}}\left(\big|\varphi_{g,3}\big\rangle_4 + \big|\varphi_{g,4}\big\rangle_4\right),\ |\Xi_4\rangle = \tfrac{1}{\sqrt{2}}\left(\big|\varphi_{g,3}\big\rangle_4 - \big|\varphi_{g,4}\big\rangle_4\right). \tag{15}$$

In fact, these results can also be obtained by another way[31]. We assume that the MES takes the form $|\Xi\rangle = k_1\big|\varphi_{g,1}\big\rangle_4 + e^{i\phi_1}k_2\big|\varphi_{g,2}\big\rangle_4 + e^{i\phi_2}k_3\big|\varphi_{g,3}\big\rangle_4 + e^{i\phi_3}k_4\big|\varphi_{g,4}\big\rangle_4$, here $k_i(i = 1, 2, 3, 4)$ represents the coefficient with $k_1^2 + k_2^2 + k_3^2 + k_4^2 = 1$. By numerically traversing the parameter spaces for $k_i(i = 1, 2, 3, 4) \in [0, 1]$ and $\varphi_j(j = 1, 2, 3) \in [-\pi, \pi]$, we can obtain the same results with those in Eq. (15). When applying electric flux operator $\widetilde{F}_x$ and magnetic flux operator $\widetilde{T}_x$ to MESs ($|\Xi_1\rangle$, …, $|\Xi_4\rangle$), the following relations are obtained,

$$
\begin{aligned}
\widetilde{F}_x|\Xi_1\rangle &= |\Xi_1\rangle, \widetilde{T}_x|\Xi_1\rangle = |\Xi_1\rangle, \widetilde{F}_x|\Xi_2\rangle = -|\Xi_2\rangle, \widetilde{T}_x|\Xi_2\rangle = |\Xi_2\rangle, \widetilde{F}_x|\Xi_3\rangle \\
&= |\Xi_3\rangle, \widetilde{T}_x|\Xi_3\rangle = -|\Xi_3\rangle, \widetilde{F}_x|\Xi_4\rangle = -|\Xi_4\rangle, \widetilde{T}_x|\Xi_4\rangle = -|\Xi_4\rangle,
\end{aligned} \tag{16}
$$

which means that there is no quasi-particle excitation in state $|\Xi_1\rangle$. Thus, we choose one MES for the square lattice with 4 spins as

$$|\psi\rangle_4|\Xi_1\rangle = \frac{1}{\sqrt{2}}\left(\big|\varphi_{g,1}\big\rangle_4 + \big|\varphi_{g,2}\big\rangle_4\right). \tag{17}$$

Eq. (17) is identical with Eq. (2). Similar to the above process, we can also obtain the MESs for the square lattices with 8 and 12 spins (details are presented in Supplementary Note 1).

**Experimental realization of 4-cebit CMES**. Here we describe the details of the process illustrated in Fig. 3b, which shows the arrangements in the DSP module to reproduce the results corresponding to those for the 4-qubit state $|\psi\rangle_4$. Eight signals in channels $\{h_i, v_i\}(i = 1, 2, 3, 4)$ are down-converted to their own frequencies as planned, and the corresponding signals output from the $\{h_i, v_i\}$ channel of the DDC are denoted by $\{B_{h_i}(t), B_{v_i}(t)\}$. Then the following procedure PROJ comprised of four units performs the desired projective operations on these signals. Each unit includes two multiplications $\left(\times c_{h_i}^*, \times c_{v_i}^*\right)$ in parallel and an adder to transform the signal pair $\{B_{h_i}(t), B_{v_i}(t)\}$ into the resultant signal $S_i(t)$. Next, the output signals $\{S_1(t), S_2(t)\}$ are sent to a multiplier ($\times$) and a digital finite-impulse-response (FIR) filters $F_{12}$, yielding a signal amplitude $S_{12}(t)$. The procedure from $\{S_3(t), S_4(t)\}$ to $S_{34}(t)$ is similar. Finally, these two summed signals $S_{12}(t)$ and $S_{34}(t)$ continue to go through mixing by a multiplier followed by a FFT-based digital filter. We record the complex amplitude of the desired output frequency signal $A_\Omega$ as described in Eq. (5), which corresponds to the projection probability amplitude of $|\psi\rangle_4$ onto a measurement basis setting. Note the modular square data $|A_\Omega|^2$ lead to similar multi-fold coincidence events registered in the quantum set-up. A more detailed description of these processes is presented in Supplementary Note 3.

## Data availability
Any related experimental background information not mentioned in the text are available from the corresponding author upon reasonable request. The source data underlying Figs. 2, 5, and Supplementary Fig. 8 are provided as a Source Data file.

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

## Acknowledgements

This work was supported by the National Key Research and Development Program of China under Grant No. 2017YFA0303800 and the National Natural Science Foundation of China (61421001, 11604014 and 61301188).

## Author contributions

T.C. finished the theoretical scheme for measuring TEE with the help of S-P.K., S.H.Z. designed the experiments and analyzed the experimental data, Y.Z. provided experimental data under the supervision of H.J.S., S.H.Z., and Y.L.L. conducted compressed sensing method for state tomography, X.D.Z. initiated and designed this research project. T.C., S.H.Z., S-P.K., and X.D.Z. wrote the manuscript.

## Additional information

**Competing interests:** The authors declare no competing interests.

