## [Peer Review File · Nature Communications]

Reviewers' comments:

Reviewer #1 (Remarks to the Author):

In this manuscript, Chen and collaborators have proposed a new way to simulate the topological order by measuring the topological entanglement entropy (TEE) based on microwave graph states (MGS). As an example, they explicitly work on the pure Kitaev toric-code model, which is exactly solvable and has Z_2 topological order in its ground state. They performed the microwave experiments following the theoretical proposal, and demonstrated experimentally that the MGS possess similar characteristics corresponding to those of the quantum cluster states. Based on the graph states, they measured the von Neumann entropy and the Z_2 topological order by calculating the TEE, which is in good agreement with the theoretical predictions. Moreover, they have extended the graph state scheme to the case with external magnetic field and experimentally simulate this field driven quantum phase transition from the Z_2 topological ordered state to the topologically trivial spin polarized state.

The results are interesting and the proposal for experimental simulation of TEE is also promising. However, I found that the current study may misunderstand some fundamental concepts of measuring TEE using the proposed scheme. The boundary of the entanglement cut between subsystem A and B in Fig. 1 is not smooth and has sharp corners. As a result, it may have significant sharp corner contribution which may depend on the number of corners and the details of the models. It is possible that for the Kitaev Toric-Code model that such a sharp corner contribution is zero, however, this needs to be verified by the authors. Even this is true for the Kitaev toric-code model, this sharp corner contribution will be finite in general, and in some cases, can be very big which can be even bigger than TEE itself. Therefore, in order to measure the TEE in a correct way, the authors need to make sure that the boundary of the entanglement cut is indeed smooth.

In addition to the key problem mentioned above, there is another issue, i.e., the representation of the toric-code model from (a1)-(c1) in Fig.1 to (a2)-(c2) is very confusing. I read this part several times, however, I still cannot figure out the exact mapping procedure, in particular, how the authors choose the sequence of the numbered sites during the mapping. For instance, the mapping between (a1) to (a2) does not allow one to directly map (b1) to (b2), so does (c1) to (c2). This is also true that the mapping from (b1) to (b2) does not allow one to directly map from (c1) to (c2). Therefore, in order to improve the readability, I suggest the authors revise this part accordingly. Moreover, the mapped graph states in (a2)-(c2) do not have smooth boundary for the entanglement cut between subsystem A and B, as shown in (a1)-(c1).

If the authors can resolve the above issues and revise the manuscript accordingly, I will be happy to recommend its publication. However, in its present form, I cannot recommend its publication since these issues are critical to validate the current results.

Reviewer #3 (Remarks to the Author):

This manuscript discusses the generation of wave analogs to the ground states of the Toric Code and the measurement of quantities that correspond to the topological entanglement entropy.

The experiments are accurately described on a technical level and I don't see any flaws in how the data is obtained. I'm not a real expert on these technical aspects of the setup though.

Where I however see significant deficits in the paper is the interpretation of the results. The authors invoke an analogy between quantum information processing and classical wave optics. Classical wave

optics allows for the observation of correlations of the same form as quantum correlations only between several degrees of freedom of the same particle, but not between degrees of freedom of different particles. The implications of this difference on how far the analogy to topological order, that is explored here, can be taken, need to be discussed thoroughly. That is differences between qubits and "cebits" need to be discussed in much greater depth.

As a start, the presentation does not explicitly say that it is an analogy to topological order that is explored, but not topological order itself. In that respect, the title is not fully appropriate to me, as it implies claims that can in my view not be made. The scheme doesn't generate real quantum correlations between different qubits, as e.g.

<https://journals.aps.org/pr/abstract/10.1103/PhysRevA.95.042330> would do.

Related to that. The authors talk about fidelities for cebits. What does 'fidelity' mean here?

A minor note:

The authors claim that their approach shows good scalability. Yet I understand that the required resources scale exponentially in the number of simulated qubits.

Reply to the Reviewer 1

Firstly we would like to thank Reviewer 1 for bringing us useful comments and suggestions to improve the manuscript. All suggestions have been implemented in the revised manuscript. Here we would like to give reply to these comments in detail.

Comment 1: *“The results are interesting and the proposal for experimental simulation of TEE is also promising. However, I found that the current study may misunderstand some fundamental concepts of measuring TEE using the proposed scheme. The boundary of the entanglement cut between subsystem A and B in Fig. 1 is not smooth and has sharp corners. As a result, it may have significant sharp corner contribution which may depend on the number of corners and the details of the models. It is possible that for the Kitaev Toric-Code model that such a sharp corner contribution is zero, however, this needs to be verified by the authors. Even this is true for the Kitaev toric-code model, this sharp corner contribution will be finite in general, and in some cases, can be very big which can be even bigger than TEE itself. Therefore, in order to measure the TEE in a correct way, the authors need to make sure that the boundary of the entanglement cut is indeed smooth.”*

Reply: In our previous manuscript, the boundary is not smooth and there indeed exists the additional contributions from sharp corner. According to the suggestion of the reviewer, we have revised the manuscript and made sure that the boundary between the subsystem A and B is smooth in the revised manuscript. Because it has been demonstrated that the toric code model is equivalent to a square lattice in terms of studying the TEE, we consider the

equivalent square lattice in the revised manuscript, where the operators σ_x and σ_z occupy the vertices instead of bounds. As presented in Fig. 1(a1)-(c1) [(a2)-(c2)] of the revised main text, we choose the torus geometries for square lattices with 4, 8 and 12 spins. To measure the TEE in a correct way, we cut the lattices into two subsystems with two disconnected smooth closed curves (see Fig. 1(a1)-(c1) [(a2)-(c2)] in the revised main text). The yellow and pink regions represent the subsystem A and B, respectively. In these cases, two disconnected boundaries between subsystem A and B are smooth, and there is no additional contribution to the entanglement entropy of system from sharp corners.

For these square lattices with torus geometries (4, 8 and 12 spins in Fig. 1(a1)-(c1) [(a2)-(c2)]), there exists four degenerated ground states. To reveal the topological properties of system, we construct the minimum entropy states (MESs) based on these degenerated ground states, and extract the topological entanglement entropy (TEE) from the entanglement entropies in these MESs. The explicit forms of MESs for the square lattices with 4, 8 and 12-spins have been addressed in Table 1 of the revised main text. In S1 of revised Supplementary Information (SI), we have provided the calculation details of obtaining these MESs. The associated experiment has been realized based on the microwave setup, and the results have been presented in Fig. 2 of the revised main text.

The corresponding results for the transition from Z_2 topological order to topologically trivial phase with the external fields have also been modified (see Fig. 5 in revised main text and related discussions). In addition, we have discussed the measurement of TEE for the toric code model in S2 and S7 of the revised SI. The MESs for the toric code model have been presented in these sections where the correspondence between the toric code model and square lattices has been demonstrated.

Comment 2: *“In addition to the key problem mentioned above, there is another issue, i.e., the representation of the toric-code model from (a1)-(c1) in Fig.1 to (a2)-(c2) is very confusing. I read this part several times, however, I still cannot figure out the exact mapping procedure, in particular, how the authors choose the sequence of the numbered sites during the mapping.*

For instance, the mapping between (a1) to (a2) does not allow one to direct map (b1) to (b2), so does (c1) to (c2). This is also true that the mapping from (b1) to (b2) does not allow on to directly map from (c1) to (c2). Therefore, in order to improve the readability, I suggest the authors revise this part accordingly. Moreover, the mapped graph states in (a2)-(c2) do not have smooth boundary for the entanglement cut between subsystem A and B, as shown in (a1)-(c1).”

Reply: In our previous manuscript, the graph states in Fig. 1(a2)-(c2) only describe the construction of ground states of toric code model in Fig. 1(a1)-(c1), respectively. Since the choice of boundary in the previous manuscript is not smooth, it can bring corner contributions to the entanglement entropy. In our revised manuscript, to make the cut between two subsystems smooth, we choose the system possessing the torus geometry (see Fig. 1(a1)-(c1) in the revised main text). In this case, the system has four degenerated ground states. Only the minimum entropy states (MESs) composed of these degenerated ground states reveals the topological entanglement entropy (TEE). In Fig. 1 of the revised main text, we have not provided the construction of these MESs as what we did in the previous manuscript. We have provided the details of construction of MESs in the revised SI (the discussion of the square lattice is addressed in S1 and S6; the study of toric code model is presented in S2 and S7.) We have also provided some explicit forms of MESs for the square lattice in the revised main text. The MESs for the square lattice without the external fields are shown in Table 1, and some MESs for the square lattice with the external fields are addressed as Eq. (6) and (7). We believe that our revision can avoid the confusion of the previous manuscript, and improve the readability.

Reply to the Reviewer 2

Firstly we would like to thank Reviewer 2 for bringing us useful comments and suggestions to improve the manuscript. All suggestions have been implemented in the revised manuscript. Here we would like to give reply to these comments in detail.

Comment 1: “ *This manuscript discusses the generation of wave analogs to the ground states*

of the Toric Code and the measurement of quantities that correspond to the topological entanglement entropy. The experiments are accurately described on a technical level and I don't see any flaws in how the data is obtained. I'm not a real expert on these technical aspects of the setup though."

"Where I however see significant deficits in the paper is the interpretation of the results. The authors invoke an analogy between quantum information processing and classical wave optics. Classical wave optics allows for the observation of correlations of the same form as quantum correlations only between several degrees of freedom of the same particle, but not between degrees of freedom of different particles. The implications of this difference on how far the analogy to topological order, that is explored here, can be taken, need to be discussed thoroughly. That is differences between qubits and "cebits" need to be discussed in much greater depth."

Reply: This is true that classical wave optics allows for the observation of correlations of the same form as quantum correlations between several degrees of freedom of the same particle. In fact, in the past years many simulations (or emulations) of quantum multi-partite systems have been performed by using multiple classical optical beams, microwaves or electronic signals with in-depth discussions of the cost of classical resources (see Refs. [39, 46, 47, 48, 49, 50] in the revised manuscript). Inspired by this area of research, in this work we have investigated how to employ a specific classical system to observe an analogy of topological phases in the toric code model and the transition from Z_2 topologically ordered state to topologically trivial phase, and thus obtained enlightening results in agreement with theoretical predictions.

However, the difference between a quantum simulator and our classical analogy is the scaling behavior of the resources required in simulating the target quantum many-body system. A well-designed genuine quantum system (e.g. superconducting quantum circuits) usually owns good scalability in terms of the resources used [58, 59], while a classical analog system does not for most cases. For example, for the case of the analogs of the 8-qubit states $|\Xi\rangle_8$ and $|\Phi_g\rangle_8$ with external fields (see S8 in Supplementary Information), each superposed term in the target state is identified with a frequency component and selected by

the use of an associated FFT-based filter followed by a modulator. Thus, the total number of filters required scales as 2^n with the qubit number n . That is to say, the whole bandwidth required for the filtering process would grow exponentially with the number of qubits in a general state to be simulated, similar to the conclusions drawn in previous work [48]. In contrast to these analogies of general states, it is also interesting to note for the analogy of certain specific states (e.g. the 4, 8, 12-qubit CMES), the amount of employed classical resources can be reduced to some extent by appropriate designs. In particular, in our design the numbers of receiving antennas, signal channels $\{h_i, v_i\}$ fed into the DSP module, and digitally down-converted signals $\{B_{hi}(t), B_{vi}(t)\}$ grow linearly with the number of cebits (see Fig.3(b), Figs. S11 and S12 in Supplementary Information), while the number of FFT-based filters at the bottom of each designed circuit in the DSP module usually depends on the number of superposed terms in the target state. In the revised manuscript, we have added these descriptions in “Discussion and Summary”.

As for the differences between qubits and "cebits", it has been discussed in Refs.[46, 47] in detail. In our system (see Fig.3), the "cebit" represents the vector form of a signal pair as the classical counterpart of a single-qubit quantum state, and these cebits form an inner product space where the inner product is given by “parentheses” $\langle \rangle$ [46, 47]. In the revised manuscript, we have emphasized such a point (see the descriptions below Eq.(2) in the main text and S3 in Supplementary Information).

Comment 2: *“As a start, the presentation does not explicitly say that it is an analogy to topological order that is explore, but not topological order itself. In that respect, the title is not fully appropriate to me, as it implies claims that can in my view not be made. The scheme doesn't generate real quantum correlations between different qubits, as e.g. <https://journals.aps.org/pr/abstract/10.1103/PhysRevA.95.042330> would do.*

Related to that. The authors talk about fidelities for cebits. What does 'fidelity' mean here?.”

Reply: According to the suggestion of the reviewer, in the revised manuscript we have added the description of classical analogy to topological order through the manuscript, including

title, abstract and main text.

As for the fidelities for cebits, we have added relevant descriptions. In the field of quantum information, the fidelity is commonly used to judge the quality of a produced state compared with the desired one. Similarly, here we use this notion to measure the degree of similarity between our experimental simulation results (e.g. $\hat{\rho}_4^{\text{cl}}$) and the target analogy state (e.g. $|\psi_4^{\text{cl}}\rangle$), which can be quantified as $(\psi_4^{\text{cl}} | \hat{\rho}_4^{\text{cl}} | \psi_4^{\text{cl}})$ for the 4-cebit case similar to those used in quantum experiments [51-56]. In this sense, we believe the reconstructed matrix with very good fidelity actually reflects good reliability of our signal processing. In the revised manuscript, we have emphasized such a point (see the descriptions above Fig.3) and added citations (see Refs. [51-56] in the revised manuscript).

Comment 3: *A minor note: “The authors claim that their approach shows good scalability. Yet I understand that the required resources scale exponentially in the number of simulated qubits.”*

Reply: Actually in our design, the numbers of receiving antennas, signal channels $\{h_i, v_i\}$ fed into the DSP module, and digitally down-converted signals $\{B_{ni}(t), B_{vi}(t)\}$ grow linearly with the number of cebits (see Fig.3(b), Figs. S11 and S12 in Supplementary Information), while the number of frequency components filtered by the final FFT-based digital filters depends on the number of corresponding superposition terms in the simulated state, and thus the number of filters or required bandwidth would scale exponentially for simulating a general state (see Fig. S19 in S8 of Supplementary Information). To avoid ambiguity, we have removed the statement about “scalability” of our experimental setup, and added detailed discussions about the applicability of our approach in the “Discussion and Summary” in the revised manuscript.

Reviewers' comments:

Reviewer #1 (Remarks to the Author):

The authors have addressed all my questions and comments in the revised manuscript, and I am happy to recommend its publication in Nature Communication.

Reviewer #2 (Remarks to the Author):

The authors have clarified my previous questions and as far as I can tell also those of other referees. Importantly, the text is now clearer about the fact that this is an analogy of topological order and entanglement in classical systems. Yet, the final sentence of the abstract still suggests that the proposed scheme would be suitable for exploring long range entanglement and topological order. In my view this is too strong a claim, even if the rest of the text is no more honest.

Apart from this aspect, I believe the paper is suitable for publication. I however strongly recommend to check the use of English language and revise as necessary.

Reviewer #2 (Remarks to the Author):

An entropic quantity was calculated for a series of states of increasing size L that are formally analogous to particular superpositions of ground states of the Kitaev toric code Hamiltonian. The states were produced using classical light beams, whose main difference from true quantum systems is that all degrees of freedom are confined to a small spatial region. Therefore, no notion of distance can be reasonably defined in such systems [47], and thus there is no traditional notion of topological order (which has to do with protecting against *local* perturbations). Nevertheless, the system admits a factorizable Hilbert space on which one constructs entangled (i.e., non-separable) states.

In the first version of the manuscript, the authors considered the toric code with open boundary conditions and picked regions A that were disk-like (i.e., not having boundaries consisting of topologically nontrivial closed loops [34]). Since the regions were quite small, non-asymptotic contributions may depend both on the region picked and the particular ground state used (the $L = 8, 14$ cases had degenerate ground states). Nevertheless, the authors created a series of *particular* ground states (graph states) whose entanglement entropy yielded the correct sub-leading term $\gamma = \ln 2$ (this can be verified quite easily via numerics).

In this version, the authors abandoned their previous approach and considered the toric code with periodic boundary conditions. They picked regions were *not* disk-like, cutting the torus into two. In such regions, the entanglement entropy is known to depend on the ground state, and this dependence was explicitly calculated in Sec. II.C.2 of [34]. The authors once again picked a series of *particular* ground states (“MES” states) whose entanglement entropy yielded the correct sub-leading term γ .

I think this is a nice experiment that likely sets records for the fidelities of such exotic $O(10)$ -qubit states. I would recommend that the authors do the following; the first point is mandatory, the latter two optional:

1. Be more careful about calling that the extracted sub-leading entanglement entropy contribution “universal”. The entanglement entropy generically contains additional non-universal terms that are of the same order as γ . So if one does not know what state one is in, one would not be able to reliably extract γ unless they tried many different states and regions. Since they cannot consider arbitrary size regions, the authors, in both versions, removed non-universal terms by fine-tuning the regions and the states. This should be more clearly discussed in the manuscript, e.g., in text around lines 92-101 and 116-119.
2. Since the authors have tomographically obtained the states, they can consider canceling L -dependence by calculating

$$I(A : C|B) = H_{AB} + H_{BC} - H_B - H_{ABC} = -2\gamma, \quad (1)$$

where H_O is the von Neumann entropy of the reduced state at O [13]. This way, they do not need a sequence of states but instead can calculate this for each size $L = 8, 12$ (with $L = 4$ being too small). For $L = 8$, the regions $A = \{6\}$, $B = \{5, 7\}$, and $C = \{8\}$. For $L = 12$, $A = \{8\}$, $B = \{7, 9, 10, 12\}$, and $C = \{11\}$.

3. I do not see any reason why they should not at least mention their first-version results with closed boundary conditions.

Reply to the Reviewer 2

Firstly we would like to thank Reviewer 2 for bringing us useful comments and suggestions to improve the manuscript. All suggestions have been implemented in the revised manuscript. Here we would like to give reply to these comments in detail.

Comment 1: *“The authors have clarified my previous questions and as far as I can tell also those of other referees. Importantly, the text is now clearer about the fact that this is an analogy of topological order and entanglement in classical systems. Yet, the final sentence of the abstract still suggests that the proposed scheme would be suitable for exploring long range entanglement and topological order. In my view this is too strong a claim, even if the rest of the text is no more honest.*

Apart from this aspect, I believe the paper is suitable for publication. I however strongly recommend to check the use of English language and revise as necessary.”

Reply: In order to avoid strong claims, in the revised manuscript we have deleted the final sentence of the abstract in the previous manuscript. The English expression has been checked and revised.

Reply to the Reviewer 3

Firstly we would like to thank Reviewer 3 for bringing us useful comments and suggestions to improve the manuscript. All suggestions have been implemented in the revised manuscript. Here we would like to give reply to these comments in detail.

Comment 1: *“Be more careful about calling that the extracted sub-leading entanglement entropy contribution “universal”. The entanglement entropy generically contains additional*

non-universal terms that are of the same order as γ . So if one does not know what state one is in, one would not be able to reliably extract γ unless they tried many different states and regions. Since they cannot consider arbitrary size regions, the authors, in both versions, removed non-universal terms by fine-tuning the regions and the states. This should be more clearly discussed in the manuscript, e.g., in text around lines 92-101 and 116-119.”

Reply: It is true that the entanglement entropy generically contains additional non-universal terms that are of the same order as topological entanglement entropy (TEE) γ . In our study of TEE for the square lattice, the non-universal term can affect the entanglement entropy for the lattice if the boundary condition or cut is not chosen appropriately. To avoid this non-universal effect, we have chosen the periodic boundary condition in the lattice and the system displays the torus geometry. We have made a non-contractible cut in the lattice to divide the system into subsystem A and B (Fig. 1(a1)-(c1) of main text). With such cut in the torus geometry, the TEE for the lattice can be extracted as a constant (not L_x -dependence) from entanglement entropy for the minimum entropy states (MESs) which are equal superposition of two degenerated ground states [34]. To illustrate such extraction of γ clearly, we have emphasized the selection of boundary, cut and MESs in Page 4 and 5 of the revised main text (see the last paragraph in Page 4 and the first paragraph in Page 5).

Comment 2: *“Since the authors have tomographically obtained the states, they can consider canceling L -dependence by calculating*

$$I(A:C|B) = H_{AB} + H_{BC} - H_B - H_{ABC} = 2\gamma$$

where H_O is the von Neumann entropy of the reduced state at O [13]. This way, they do not need a sequence of states but instead can calculate this for each size $L = 8; 12$ (with $L = 4$ being too small). For $L = 8$, the regions $A = \{6\}$, $B = \{5, 7\}$, and $C = \{8\}$. For $L = 12$, $A = \{8\}$, $B = \{7, 9, 10, 12\}$ and $C = \{11\}$.”

Reply: We thank the reviewer’s advice on calculation of quantum conditional mutual

information $I(A:C|B)$. This quantity describes the tripartite correlation in which the L -dependent parts are all cancelled out. We have chosen the regions A, B and C in the 8-spins and 12-spins system following the reviewer's instruction. Our theoretical calculations have shown that when the lattice has 8 or 12 spins, the quantity $I(A:C|B)$ is indeed 2γ . With the experimentally obtained states, we also find that this quantity $I(A:C|B)$ is very close to 2γ .

Comment 3: *"I do not see any reason why they should not at least mention their first-version results with closed boundary conditions."*

Reply: For our first-version results, the cut in the system is not smooth. Therefore, the corner may have the contribution to the entanglement entropy. Following the suggestion of Reviewer 1, in the second-version manuscript, we have divided the system into subsystem A and B with a smooth cut. In the main text of our revised manuscript (the third-version), we have also added the description on how to extract γ from entanglement entropy for MESs.